# Premature utilization of MRI in the workup of mild hyperprolactinemia

Jennifer A. Mann[1]*, Jared Galloway[2], Sana Ghaznavi[2], David J. T. Campbell[2], John Lysack[3], Kirstie Lithgow[2,4]

**1** Department of Clinical Neurosciences, Section of Neurosurgery, University of Calgary, Calgary, Alberta, Canada, **2** Division of Endocrinology and Metabolism, Department of Medicine, University of Calgary, Calgary, Alberta, Canada, **3** Department of Radiology, University of Calgary, Calgary, Alberta, Canada, **4** Hotchkiss Brain Institute, University of Calgary, Calgary, Alberta, Canada

* jennifer.mann@ucalgary.ca

## Abstract

### Background

Hyperprolactinemia is a common biochemical finding detected in primary care, and guidelines suggest confirmation with a second prolactin measurement to rule out spurious elevation prior to further workup. Premature magnetic resonance imaging (MRI) is a costly potential consequence of improperly worked-up hyperprolactinemia. We aimed to quantify the proportion of prematurely ordered MRI for isolated hyperprolactinemia without confirmation.

### Methods

A retrospective chart review was performed between 2012–2021 in Alberta, Canada. We identified cases of mildly elevated serum prolactin (<100mcg/L, females ≥25.0mcg/L, males ≥21.0mcg/L) in patients >18 years who underwent MRI brain or sella following prolactin measurement. MRI was classified as "clinically appropriate" or "premature" based on clinical indication. Clinically appropriate and premature groups were compared based on patient age, sex, and scan year.

### Results

We identified 1,730 cases meeting inclusion criteria with a single prolactin measurement prior to MRI; over half (56.1%, n=970) were categorized as "premature." Of the MRIs in the premature cohort, 39.4% (n=382) reported at least one abnormal finding, most commonly microadenoma (n=200). There was no statistically significant difference between premature and clinically appropriate groups for patient sex (p=0.12) or year of MRI (p=0.35), but patients in the premature group were significantly older (p<0.01, mean difference 1.52 years, 95% CI:[0.45,2.49]).

**Data availability statement:** All relevant data are within the paper and its Supporting Information files.

**Funding:** The author(s) received no specific funding for this work.

**Competing interests:** The authors have declared that no competing interests exist.

**Abbreviations:** CT, computed tomography; EMR, electronic medical record; hCG, human chorionic gonadotropin; MRI, magnetic resonance imaging; TSH, thyroid-stimulating hormone.

## Conclusions

MRI for investigation of mild hyperprolactinemia is often ordered prematurely without confirmation by repeat prolactin level and generates incidental findings of uncertain clinical significance to the presenting issue. To reduce patient stress and healthcare system burden, providers should ensure proper workup of hyperprolactinemia with repeat prolactin measurement prior to ordering MRI.

## Background

Elevated serum prolactin (hyperprolactinemia) is a biochemical finding with a broad differential diagnosis [1–3]. Hyperprolactinemia may be detected in the primary care setting during workup of amenorrhea, galactorrhea, hypogonadism, or non-specific complaints such as fatigue, and is a common reason for referral to endocrinology [2,4]. After an initial elevated prolactin level is detected, a stepwise workup should be undertaken to avoid incorrect diagnoses and/or unnecessary investigations [5]. At minimum, the presence of hyperprolactinemia should always be confirmed with a second serum prolactin measurement, as spurious elevations in prolactin can occur due to venipuncture stress, sexual intercourse, nipple stimulation, or pulsatile secretion [2,4,6]. Once hyperprolactinemia has been confirmed on two occasions, further work-up of hyperprolactinemia including physiologic, pharmacologic, and pathologic causes can be explored [2,4].

Though a single elevation of prolactin may be sufficient to refer for MRI sella in select circumstances (i.e., prolactin >500 mcg/L [4,7]), mild elevations in prolactin (<100 mcg/L) are common [6,8]. Mild elevations in prolactin can often be spurious (i.e., due to exercise, nipple stimulation, venipuncture) [4] or due to non-endocrine causes (i.e., medications, renal disease, primary hypothyroidism) [4,9]. An association with polycystic ovarian syndrome (PCOS) has been suggested by some studies [10], but evidence on this matter is conflicting [11]. Anti-psychotics in particular are a common cause of mild or hyperprolactinemia which can be seen with risperidone, olanzapine, quetiapine, and paliperidone [2]. Workup for hyperprolactinemia should include a careful and detailed history, medication review, and physical exam to exclude pharmacologic or non-endocrine causes [2]. In cases of mild, asymptomatic hyperprolactinemia, macroprolactinemia should be excluded using polyethylene glycol precipitation prior to ordering neuroimaging [2]. Repeat prolactin measurement is an important step in mild hyperprolactinemia (in the absence of red flag symptoms) to rule out spurious elevation [12]. Failure to repeat prolactin measurement in cases of mild hyperprolactinemia and premature ordering of MRI sella may lead to identification of non-functioning microadenoma, which is a common incidental finding [13,14]. A previous study of non-functioning microadenomas reported that almost 10% of cases were detected due to transient hyperprolactinemia [15]. Other incidental findings on MRI include empty sella, pituitary enlargement, and Rathke's cleft cyst [14]. Such findings are likely to generate further investigation, imaging surveillance, subspecialty appointments, and importantly, patient distress [16]. Incomplete workup of

hyperprolactinemia prior to MRI has important implications for patients, providers, and utilization of healthcare resources. The aim of our study was to quantify the proportion of MRI sella ordered prematurely for isolated hyperprolactinemia without confirmation with repeat prolactin measurement.

## Materials and methods

A retrospective chart review was performed between 2012 and 2021 in the province of Alberta, Canada. Our protocol was approved by the Conjoint Health Research Ethics Board at the University of Calgary (REB22-0273). Potentially eligible cases were identified by a data analyst from Alberta Health Services. In order to allow us to clarify clinical details from each patient chart, we received the raw data with patient identifiers. The need for informed consent was waived by our institutional research ethics board due to the large number of cases. Each case was screened using the following criteria. Inclusion criteria were: 1) patients >18 years of age; 2) mildly elevated serum prolactin >25.0 for females and >21.0 for males but <100 mcg/L; 3) MRI sella or MRI brain performed *following* the detection of hyperprolactinemia and ordered *within 1 year* of the prolactin measurement. Exclusion criteria were: 1) elevated serum prolactin >100 mcg/L; 2) cases with neuroimaging performed for alternative indications; 3) follow-up scans for pre-existing sellar masses or structural abnormalities. Cases that met initial eligibility criteria were assessed for presence or absence of repeat prolactin measurement prior to the MRI and cases with two or more prolactin measurements were excluded. In the province of Alberta, Canada, a centralized Electronic Health Record (Netcare) contains laboratory investigations, allowing us to reliably assess the number of investigations completed for each patient regardless of laboratory utilized in the province. Data was accessed from 01/07/2023 to 30/06/2024. For cases with a single prolactin measurement, we then classified MRI ordering as "clinically appropriate" or "premature" based on the clinical indication extracted from the MRI requisition. Indications classified as "clinically appropriate" included amenorrhea/hypogonadal symptoms, galactorrhea, hypopituitarism, or mass effect, as these signs and symptoms are suggestive of stalk effect or direct compressive impact of a large macroadenoma or other sellar mass [4]. We collected further data for all cases in the "premature" group including imaging findings and name of ordering provider (which was cross-referenced with the College of Physicians and Surgeons of Alberta's physician directory [17] to collect information about ordering provider speciality and practice location) with the rationale of informing future quality improvement initiatives.

Statistical analysis was performed using Stata Version 17. We compared the "clinically appropriate" and "premature" imaging groups based on patient age, sex, and year of MRI. As our study traversed the years of the global COVID-19 pandemic, we felt it necessary to explore ordering patterns by year of MRI in case these were impacted by changes to patient care resulting from the pandemic. Furthermore, we were interested in how temporal changes in the availability of healthcare resources (i.e., increased imaging wait times for routine diagnostic imaging) impacted MRI ordering practices. Binary variables were assessed using a chi-squared test and continuous variables were assessed using a two-sample t-test. Statistical significance was set at $p < 0.05$.

## Results

We identified 5,740 cases with MRI sella or brain and mildly elevated prolactin (<100 mcg/L) ordered prior to MRI between 2012 and 2021. We screened each case for further eligibility which left 3,715 cases. Of these, 46.6% (1,730) had a single prolactin measured prior to MRI (Fig 1). After further categorization based on the MRI requisition, 56.1% (n = 970) were categorized as "premature" while the remaining 43.9% (n = 760) were categorized as "clinically appropriate."

The included cohort was 84.0% (N = 1454) female, with a mean age of 35.5 years (SD 11.25; min:18 years, max:100 years) (Table 1).

Indications for MRI within the "clinically appropriate" group are summarized in Table 1 and include: amenorrhea/hypogonadal symptoms (65.7%, n = 499), galactorrhea (25.0%, n = 190), mass effect (8.0%, n = 61), and hypopituitarism (1.3%, n = 10). Indications within the "premature" group included hyperprolactinemia 70.8% (n = 687), headaches 23.5% (n = 228),

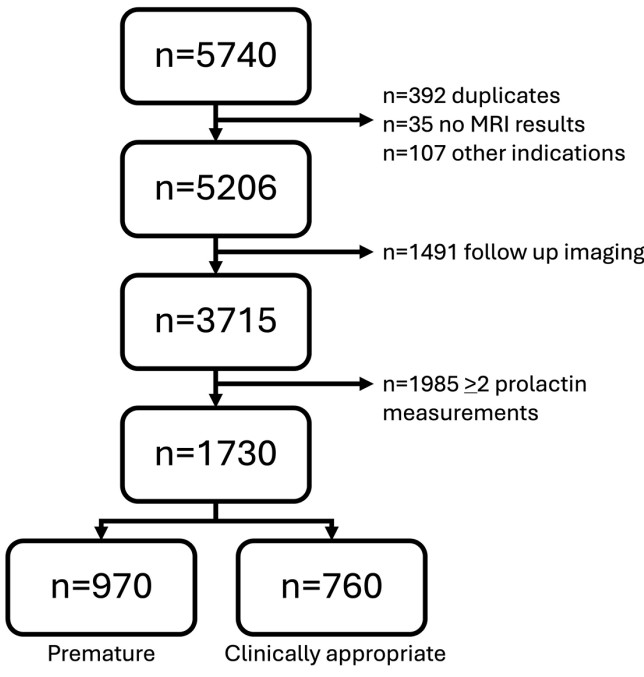

**Fig 1. Flow diagram of case inclusion and categorization.**

elevated TSH 4.4% (n = 43), and other 1.2%.(12) Of the imaging results for the "premature" group, the majority (60.6%, n = 588) reported normal findings, and 39.4% (n = 382) reported *at least one abnormal finding* (Fig 2). Most commonly, abnormal findings related to the pituitary gland or stalk and included microadenoma (20.6%, n = 200), macroadenoma (2.2%, n = 21), empty sella (1.2%, n = 12), and other pituitary abnormalities (i.e., hyperintensity on T2, asymmetry of the pituitary gland, thickened infundibulum) in 4.8% (n = 47). Non-specific white matter changes were another commonly reported finding (4.3%, n = 42).

There was no statistically significant difference between patient sex (p = 0.12) or year of the MRI (p = 0.35) (Table 1) when comparing the "premature" and "clinically appropriate" groups. Patients in the "premature" group were statistically significantly older than those in the clinically appropriate group (p = 0.0053, mean difference 1.52 years, 95% CI: [0.45,2.49]).

We analyzed information regarding ordering providers from the "premature" group as another way to inform quality improvement initiatives. The practice location was 90.1% urban, 8.9% rural, and 1.0% unknown. The majority of ordering physicians were family medicine (73.1%), obstetrics (10.8%), endocrinology (8.1%), and other (7.9%). There were 970 premature MRI's ordered by 630 unique ordering physicians. Of these 630 physicians, most (75.7%, n = 477) ordered only one MRI during the study interval, while the remaining 24.3% (n = 153) ordered multiple (mean 3.2; range: 2–16) MRIs. Additionally, there were a small number of physicians who were outliers in terms of MRI ordering; the top 2.5th percentile of ordering providers was composed of 16 physicians: 8 family physicians, 5 obstetricians, and 3 endocrinologists. These 16 physicians ordered a total of 130 MRI scans among them (comprising 13.4% of all MRIs).

## Discussion

We identified a large proportion of MRIs in our health region ordered for isolated, mild hyperprolactinemia that were performed prematurely without confirmation by repeat prolactin measurement. Initial work-up of mild hyperprolactinemia

**Table 1. Characteristics of patients who received MRI for a single prolactin measurement.**

| Characteristic | Total (n=1730) | Premature (n=970) | Clinically appropriate (n=760) | P-value |
|---|---|---|---|---|
| Mean age, years | 35.5 (11.3) | 36.2 (11.7) | 34.7 (10.7) | 0.0053* |
| Sex | | | | 0.12 |
| Male | 276 (16.0%) | 143 (14.7%) | 133 (17.5%) | |
| Female | 1454 (84.0%) | 827 (85.3%) | 627 (82.5%) | |
| Indication for MRI | | | | ------- |
| Hypogonadal symptoms | 499 | 0 | 499 (65.6%) | |
| Galactorrhea | 190 | 0 | 190 (25.0%) | |
| Mass effect | 61 | 0 | 61 (8.0%) | |
| Hypopituitarism | 10 | 0 | 10 (1.3%) | |
| Hyperprolactinemia | 687 | 687 (70.8%) | 0 | |
| Headaches | 228 | 228 (23.5%) | 0 | |
| Elevated TSH | 43 | 43 (4.4%) | 0 | |
| Other | 12 | 12 (1.2%) | 0 | |
| Year of MRI | | | | 0.35 |
| 2012 | 77 | 47 | 30 | |
| 2013 | 197 | 113 | 84 | |
| 2014 | 217 | 125 | 92 | |
| 2015 | 204 | 112 | 92 | |
| 2016 | 204 | 124 | 80 | |
| 2017 | 183 | 105 | 78 | |
| 2018 | 146 | 79 | 67 | |
| 2019 | 144 | 67 | 77 | |
| 2020 | 150 | 76 | 74 | |
| 2021 | 199 | 117 | 82 | |
| 2022 | 9 | 5 | 4 | |

Values are given as number of patients (%), mean (SD), or median [IQR] unless otherwise indicated.

* Denotes statistical significance at p<0.05

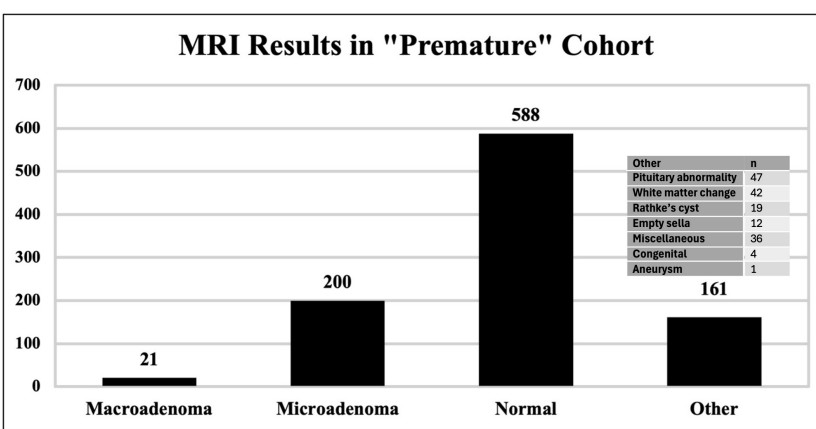

**Fig 2. Findings from "premature" MRIs.**

should include measurements of macroprolactin, beta hCG, and thyroid stimulating hormone, [5] which we did not assess, therefore our findings likely do not illustrate the full scope of "premature" imaging for this indication. We found that "premature" imaging was associated with a high prevalence (39.4%) of MRI findings of questionable clinical significance. Pituitary microadenomas were a common imaging finding, and while some may represent true microprolactinomas, most are likely incidental non-functioning microadenomas (microprolactinomas are typically associated with prolactin >100 mcg/L) [2]. With respect to interpreting abnormal imaging findings, it is important recognize that sellar neoplasms associated with prolactin < 100 mcg/L are more likely to be non-functioning pituitary adenomas, craniopharyngiomas, or Rathke's cleft cysts rather than prolactinomas [2,18]. This is in keeping with reports from other studies; a previous study of 459 non-functioning pituitary microadenomas reported that transient hyperprolactinemia was the indication for imaging in almost 10% of cases [15]. Abnormal MRI findings are likely to lead to further downstream investigations, referrals, and subspecialty appointments. For instance, guidelines recommend that pituitary microadenomas undergo surveillance MRI annually for three years [16]. This creates potentially undue burden to patients, providers, and the healthcare system.

When examining the ordering patterns of "premature" MRIs in our cohort, we found 90.1% of ordering physicians practiced in an urban environment, and 8.9% practiced in a rural environment (1.0% unknown). We emphasize that no firm conclusions can be drawn from these findings as we do not have comparative data from the "clinically appropriate" group. Extrapolation from 2021 provincial data shows 87–98% of Albertan physicians practiced in an urban environment and 2−12% practiced rurally, [19] therefore, the ordering patterns observed in this study likely mirror physician practice location.

Our findings represent a feasible and actionable target to meaningfully impact healthcare resource utilization. Possible quality improvement initiatives could be provider or systems-level focused. A previous study from Australia implemented a feedback program and decision support tool for providers with the aim of reducing unnecessary diagnostic imaging for non-specific acute low back pain [20]. This program led to a statistically significant reduction in the volume of diagnostic imaging (10.85% relative reduction in lumbosacral CT scan volume) and was highly cost effective (estimated cost reduction $11.6 million AUD) [20]. A similarly provider-focused intervention could be impactful in our setting given most "premature" MRIs were ordered by a small group of recurring providers, and a small number of physicians accounted for a high proportion of imaging. Another previous study conducted in our province [21] aimed to reduce lab testing for serum total 25-hydroxyvitamin D through implementing a requisition that allowed this testing only for specific clinical indications. This intervention led to a 91% reduction in ordering of vitamin D testing in the following 12 months and generated estimated cost savings of up to $1.5 million CAD annually [21]. As all MRI scans in our province are now protocoled through a centralized electronic medical record (EMR), a systems-level intervention requiring further diagnostic workup and/or a specific clinical indication for MRIs ordered for prolactin <100 mcg/L is potentially feasible.

Our study has several limitations. We identified MRIs that were ordered prematurely and could have potentially been avoided, however, for some cases this imaging would have ultimately proved clinically appropriate and therefore our analysis does not allow us to estimate the burden of imaging that was truly unnecessary. Cases were categorized based on available clinical information including information provided on the MRI requisition, labs, and specialist consult letters. However, due to the multitude of different EMRs utilized across our province, we were unable to access provider notes from many ordering physicians (notably family physicians), therefore the data we accessed may not have provided the full clinical context. There was a small number of cases (n = 35) for which MRI reports were unavailable (i.e., not uploaded to centralized EMR), however, given these represent only 0.6% of the total MRIs performed, this is unlikely to have meaningfully impacted our results. Finally, we could only obtain data from imaging studies performed in public healthcare facilities. Although Canada has a publicly funded healthcare system, MRI services in our province are also available at private facilities for patients willing to pay out of pocket. As such, a very small proportion of MRIs performed for this indication at private facilities were not captured by our study.

## Conclusions

MRI sella for investigation of mild hyperprolactinemia is commonly ordered prematurely without confirmation by repeat prolactin level. In our province, premature imaging generated *hundreds* of findings of questionable clinical significance (382 over the span of nine years), many of which were probably clinical insignificant non-functioning pituitary microadenomas. A large proportion of these imaging studies (and associated downstream testing and follow-up) could likely have been avoided; further study is required to characterize the true burden of unnecessary imaging and scope of subsequent effects. Future work should focus on implementation of interventions aimed at reducing premature neuroimaging for mild hyperprolactinemia targeted at both provider education and system-level interventions.

## Supporting information

**S1 File. File containing raw dataset.**
(XLSX)

## Author contributions

**Conceptualization:** Kirstie Lithgow, Jennifer A. Mann, Sana Ghaznavi, David J.T. Campbell, John Lysack.

**Data curation:** Kirstie Lithgow, Jennifer A. Mann, Jared Galloway.

**Formal analysis:** Kirstie Lithgow, Jennifer A. Mann, Jared Galloway.

**Investigation:** Kirstie Lithgow, Jared Galloway.

**Methodology:** Kirstie Lithgow, David J.T. Campbell, John Lysack.

**Resources:** Kirstie Lithgow.

**Supervision:** Kirstie Lithgow, Sana Ghaznavi, David J.T. Campbell, John Lysack.

**Writing – original draft:** Kirstie Lithgow, Jared Galloway.

**Writing – review & editing:** Kirstie Lithgow, Jennifer A. Mann, Sana Ghaznavi, David J.T. Campbell, John Lysack.

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
