## [Decision Letter · Decision Letter 0]

18 Feb 2026

PONE-D-25-64481Premature utilization of MRI in the workup of mild hyperprolactinemiaPLOS One

Dear Dr. Lithgow,

Thank you for submitting your manuscript to PLOS ONE. After careful consideration, we feel that it has merit but does not fully meet PLOS ONE’s publication criteria as it currently stands. Therefore, we invite you to submit a revised version of the manuscript that addresses the points raised during the review process.

We look forward to receiving your revised manuscript.

Kind regards,

Melissa Orlandin Premaor, M.D., Ph.D

Academic Editor

PLOS One

Journal Requirements:

2. We noted in your submission details that a portion of your manuscript may have been presented or published elsewhere. “Yes-- abstract published in Endocrine Abstracts due to presentation at European Society of Endocrinology meeting in 2024”. Please clarify whether this [conference proceeding or publication] was peer-reviewed and formally published. If this work was previously peer-reviewed and published, in the cover letter please provide the reason that this work does not constitute dual publication and should be included in the current manuscript

4. Please amend your authorship list in your manuscript file to include author Jenna Mann.

Reviewers' comments:

Reviewer's Responses to Questions

**Comments to the Author**

1. Is the manuscript technically sound, and do the data support the conclusions?

Reviewer #1: Yes

2. Has the statistical analysis been performed appropriately and rigorously?

Reviewer #1: Yes

3. Have the authors made all data underlying the findings in their manuscript fully available?

Reviewer #1: Yes

4. Is the manuscript presented in an intelligible fashion and written in standard English?

Reviewer #1: Yes

5. Review Comments to the Author

Reviewer #1: I congratulate the authors on this rigorous study addressing a critical gap in clinical practice regarding the overuse of neuroimaging techniques in patients with mild hyperprolactinemia.

I find the authors' focus on the lack of biochemical confirmation before proceeding to costly and often misleading MRI scans to be highly relevant. The finding that 56.1% of MRIs were ordered prematurely highlights a significant healthcare burden and the risk of managing clinically insignificant incidentalomas.

Suggestions for Revision

1. The authors correctly state that guidelines require a second measurement to rule out spurious elevations. In the pediatric and adolescent population, we emphasize that prolactin should be measured at least on two separate occasions to avoid misdiagnosis due to stress or pulsatile secretion. I suggest the authors further emphasize that mild elevations (<100 mcg/L) are frequently transient or related to non-pituitary factors.

2. The Role of PCOS and Medications: The manuscript would benefit from a more detailed discussion on the common etiologies of mild hyperprolactinemia that must be excluded before MRI. Specifically, Polycystic Ovary Syndrome (PCOS) and antipsychotic medications (e.g., risperidone) are leading causes of mild-to-moderate prolactin elevation. Addressing these would strengthen the clinical rationale for why an immediate MRI is often "premature".

3. Macroprolactinemia Screening: A common pitfall in hyperprolactinemia workup is neglecting macroprolactinemia, which accounts for a significant portion of "biochemical" hyperprolactinemia cases. I recommend suggesting that polyethylene glycol (PEG) precipitation should be a standard step before neuroimaging for all mild, asymptomatic cases.

4. The "Stalk Effect" and Pseudoprolactinomas: For the "premature" group with abnormal findings, it is essential to distinguish between true prolactinomas and pseudoprolactinomas (non-functioning masses causing stalk effect). In cases where prolactin is <100 mcg/L but a mass is present, the elevation is often due to impaired dopaminergic inhibition rather than a prolactin-secreting tumor.

5. To provide a broader etiological perspective and support the management of mild hyperprolactinemia in younger populations, I recommend the following recent studies be cited in the Discussion section:

For the Discussion on Etiological Spectrum (especially PCOS):"While pituitary tumors are a known cause, recent evidence in younger cohorts suggests that PCOS is a leading etiology of mild hyperprolactinemia, often showing a significant correlation with BMI-SDS. Incorporating this into the primary workup could significantly reduce the number of unnecessary MRIs." (Reference: Kilci F, et al. Hyperprolactinemia in children and adolescents: clinical characteristics and etiological spectrum. Eur J Pediatr. 2025.)

For the Discussion on Pituitary Masses and Prolactin Thresholds: "In cases where MRI is performed, it is crucial to recognize that masses associated with prolactin levels below 100 ng/mL are frequently non-functioning pituitary adenomas (NFPAs) or craniopharyngiomas rather than true prolactinomas. These findings support the authors' conclusion that mild elevations rarely predict significant prolactin-secreting tumors." (Reference: Kilci F, et al. Etiology, presentation, and outcomes of hyperprolactinemia due to pituitary masses in children and adolescents. Endocrine. 2025.)

6. PLOS authors have the option to publish the peer review history of their article (what does this mean?). If published, this will include your full peer review and any attached files.

Reviewer #1: No

---

## [Author Response · Author response to Decision Letter 1]

27 Feb 2026

Thank you, this has been done.

2. We noted in your submission details that a portion of your manuscript may have been presented or published elsewhere. “Yes-- abstract published in Endocrine Abstracts due to presentation at European Society of Endocrinology meeting in 2024”. Please clarify whether this [conference proceeding or publication] was peer-reviewed and formally published. If this work was previously peer-reviewed and published, in the cover letter please provide the reason that this work does not constitute dual publication and should be included in the current manuscript.

A preliminary version of the analysis was submitted to the European Society of Endocrinology annual meeting in 2024 and presented as a poster. All accepted abstracts to that meeting are published in the journal Endocrine Abstracts, which functions as a searchable abstract tool/record from the conference. Many abstracts published here go on to be published in full in other journals. The purpose of presenting at this meeting was to share our preliminary analysis (in poster form) and get feedback from experts in the field which contributed to our final manuscript. The full analysis and other details in our manuscript have not been published elsewhere.

Thank you, we have uploaded the data as a supporting file.

4. Please amend your authorship list in your manuscript file to include author Jenna Mann.

Full name is Jennifer Mann, which has been included.

Reviewers' comments:

Reviewer's Responses to Questions

Comments to the Author

1. Is the manuscript technically sound, and do the data support the conclusions?

Reviewer #1: Yes

2. Has the statistical analysis been performed appropriately and rigorously?

Reviewer #1: Yes

3. Have the authors made all data underlying the findings in their manuscript fully available?

Reviewer #1: Yes

4. Is the manuscript presented in an intelligible fashion and written in standard English?

Reviewer #1: Yes

5. Review Comments to the Author

Reviewer #1: I congratulate the authors on this rigorous study addressing a critical gap in clinical practice regarding the overuse of neuroimaging techniques in patients with mild hyperprolactinemia.

I find the authors' focus on the lack of biochemical confirmation before proceeding to costly and often misleading MRI scans to be highly relevant. The finding that 56.1% of MRIs were ordered prematurely highlights a significant healthcare burden and the risk of managing clinically insignificant incidentalomas.

Suggestions for Revision

1. The authors correctly state that guidelines require a second measurement to rule out spurious elevations. In the pediatric and adolescent population, we emphasize that prolactin should be measured at least on two separate occasions to avoid misdiagnosis due to stress or pulsatile secretion. I suggest the authors further emphasize that mild elevations (<100 mcg/L) are frequently transient or related to non-pituitary factors.

Thank you for your thoughtful and detailed review of our manuscript. We agree with further emphasizing this point and have revised the introduction (lines 72- to 81) accordingly.

2. The Role of PCOS and Medications: The manuscript would benefit from a more detailed discussion on the common etiologies of mild hyperprolactinemia that must be excluded before MRI. Specifically, Polycystic Ovary Syndrome (PCOS) and antipsychotic medications (e.g., risperidone) are leading causes of mild-to-moderate prolactin elevation. Addressing these would strengthen the clinical rationale for why an immediate MRI is often "premature".

Thank you for this comment. The association between PCOS and hyperprolactinemia is controversial. While an association has been observed, and a clear mechanistic, causative link has not been established. A recent cross sectional study suggested that PCOS patients (n=1429) do not have higher prolactin levels than controls (1.3% vs 3%, p=0.05) (van der Ham et al Front Endocrinol 2023 Oct 3:14:1245106). Due to uncertainty on this topic, we feel patients with hyperprolactinemia and a concurrent diagnosis of PCOS should still undergo workup for potential alternative etiologies. However, we have added a line indicated that PCOS has been associated with hyperprolactinemia in some studies. We have added additional information about antipsychotic use in in lines 76 to 78.

3. Macroprolactinemia Screening: A common pitfall in hyperprolactinemia workup is neglecting macroprolactinemia, which accounts for a significant portion of "biochemical" hyperprolactinemia cases. I recommend suggesting that polyethylene glycol (PEG) precipitation should be a standard step before neuroimaging for all mild, asymptomatic cases.

This has been added (lines 81 to 83).

4. The "Stalk Effect" and Pseudoprolactinomas: For the "premature" group with abnormal findings, it is essential to distinguish between true prolactinomas and pseudoprolactinomas (non-functioning masses causing stalk effect). In cases where prolactin is <100 mcg/L but a mass is present, the elevation is often due to impaired dopaminergic inhibition rather than a prolactin-secreting tumor.

Unfortunately, we were not able to access consultation notes through our EMR review (many endocrinologists and other subspecialists work in community clinics outside our academic centre and clinical notes are not reported in our hospital EMR, unlike labs and imaging which are centralized). Presumably, many patients would have undergone endocrinology referral and/or further work-up of hyperprolactinemia to determine etiology (stalk effect vs. prolactin hypersecretion vs. other etiology), but we are not able to specifically report on this.

5. To provide a broader etiological perspective and support the management of mild hyperprolactinemia in younger populations, I recommend the following recent studies be cited in the Discussion section:

For the Discussion on Etiological Spectrum (especially PCOS):"While pituitary tumors are a known cause, recent evidence in younger cohorts suggests that PCOS is a leading etiology of mild hyperprolactinemia, often showing a significant correlation with BMI-SDS. Incorporating this into the primary workup could significantly reduce the number of unnecessary MRIs." (Reference: Kilci F, et al. Hyperprolactinemia in children and adolescents: clinical characteristics and etiological spectrum. Eur J Pediatr. 2025.)

Please see our response to #2 above. While we fully acknowledge that this study and others have shown an association of PCOS and mild hyperprolactinemia, a causative mechanism has not been established and evidence on this matter is conflicting. Therefore, it is imperative that these patients still undergo workup for other potential etiologies. We have added the suggested citation with another showing conflicting results to demonstrate the equipoise on this matter.

For the Discussion on Pituitary Masses and Prolactin Thresholds: "In cases where MRI is performed, it is crucial to recognize that masses associated with prolactin levels below 100 ng/mL are frequently non-functioning pituitary adenomas (NFPAs) or craniopharyngiomas rather than true prolactinomas. These findings support the authors' conclusion that mild elevations rarely predict significant prolactin-secreting tumors." (Reference: Kilci F, et al. Etiology, presentation, and outcomes of hyperprolactinemia due to pituitary masses in children and adolescents. Endocrine. 2025.)

This has been added (lines 95 to 98)

6. PLOS authors have the option to publish the peer review history of their article (what does this mean?). If published, this will include your full peer review and any attached files.

Do you want your identity to be public for this peer review? For information about this choice, including consent withdrawal, please see our Privacy Policy.

Reviewer #1: No

---

## [Editor Report · Decision Letter 1]

16 Mar 2026

Premature utilization of MRI in the workup of mild hyperprolactinemia

PONE-D-25-64481R1

Dear Dr. Lithgow,

We’re pleased to inform you that your manuscript has been judged scientifically suitable for publication and will be formally accepted for publication once it meets all outstanding technical requirements.

Kind regards,

Melissa Orlandin Premaor, M.D., Ph.D

Academic Editor

PLOS One
---

## [Editor Report · Acceptance letter]

PONE-D-25-64481R1

PLOS One

Dear Dr. Lithgow,

I'm pleased to inform you that your manuscript has been deemed suitable for publication in PLOS One. Congratulations! Your manuscript is now being handed over to our production team.

Kind regards,

on behalf of

Dr. Melissa Orlandin Premaor

Academic Editor

PLOS One